# Gut Microbial Disruption in Critically Ill Patients with COVID-19-Associated Pulmonary Aspergillosis

**DOI:** 10.3390/jof8121265

**Published:** 2022-11-30

**Authors:** H. Carlo Maurer, David Schult, Plamena Koyumdzhieva, Sandra Reitmeier, Moritz Middelhoff, Sebastian Rasch, Markus List, Klaus-Peter Janssen, Katja Steiger, Ulrike Protzer, Roland M. Schmid, Klaus Neuhaus, Dirk Haller, Michael Quante, Tobias Lahmer

**Affiliations:** 1Department of Internal Medicine II, Klinikum Rechts der Isar, School of Medicine, Technical University of Munich, 81675 Munich, Germany; 2ZIEL Institute for Food & Health, School of Life Sciences, Technical University of Munich, 85354 Freising, Germany; 3Chair of Nutrition and Immunology, School of Life Sciences, Technical University of Munich, 85354 Freising, Germany; 4Chair of Experimental Bioinformatics, School of Life Sciences, Technical University of Munich, 85354 Freising, Germany; 5Department of Surgery, Klinikum Rechts der Isar, School of Medicine, Technical University of Munich, 81675 Munich, Germany; 6Institute of Pathology, School of Medicine, Technical University of Munich, 81675 Munich, Germany; 7Institute of Virology, School of Medicine and Helmholtz Zentrum Munich, Technical University of Munich, 81675 Munich, Germany; 8Department of Internal Medicine II, School of Medicine, University Hospital of Freiburg, 79106 Freiburg, Germany

**Keywords:** COVID-19, aspergillosis, CAPA, microbiome, gut

## Abstract

Objectives: COVID-19 disease can be exacerbated by *Aspergillus* superinfection (CAPA). However, the causes of CAPA are not yet fully understood. Recently, alterations in the gut microbiome have been associated with a more complicated and severe disease course in COVID-19 patients, most likely due to immunological mechanisms. The aim of this study was to investigate a potential association between severe CAPA and alterations in the gut and bronchial microbial composition. Methods: We performed 16S rRNA gene amplicon sequencing of stool and bronchial samples from a total of 16 COVID-19 patients with CAPA and 26 patients without CAPA. All patients were admitted to the intensive care unit. Results were carefully tested for potentially confounding influences on the microbiome during hospitalization. Results: We found that late in COVID-19 disease, CAPA patients exhibited a trend towards reduced gut microbial diversity. Furthermore, late-stage patients with CAPA superinfection exhibited an increased abundance of *Staphylococcus epidermidis* in the gut which was not found in late non-CAPA cases or early in the disease. The analysis of bronchial samples did not yield significant results. Conclusions: This is the first study showing that alterations in the gut microbiome accompany severe CAPA and possibly influence the host’s immunological response. In particular, an increase in *Staphylococcus epidermidis* in the intestine could be of importance.

## 1. Introduction

Super- and/or co-infections are well-known complications that arise in critically ill patients with severe acute respiratory syndrome coronavirus type 2 (SARS-CoV-2) pneumonia [1,2,3].

Similar to patients suffering from severe influenza pneumonia, coronavirus disease 19 (COVID-19) patients may develop invasive pulmonary aspergillosis during their treatment in the intensive care unit (ICU) [4,5]. COVID-19-associated pulmonary aspergillosis (CAPA) is an opportunistic secondary infection primarily affecting ICU patients with severe acute respiratory distress syndrome (ARDS) [5], which increases both morbidity and mortality [6,7]. Besides immunological mechanisms caused by SARS-CoV-2 infection, invasive mechanical ventilation, concomitant corticosteroid or antibody treatment, and older age, were identified as independent risk factors for increased susceptibility to invasive fungal infections [6,7].

Changes in the gut microbiome are known to impact the immune response via numerous interactions between intestinal bacteria and mucosal immune cells [8,9]. Recent studies indicate that not only global changes in the gut microbial composition but also certain individual bacteria may contribute to the inflammatory response and disease severity in COVID-19 patients [10,11,12,13]. Lately, our group has shown that changes in intestinal bacterial composition are associated with an increased complication rate of COVID-19 [13]. Building on these results, we examined the association of gut and bronchial microbial composition and the risk of developing CAPA in a cohort of critically ill COVID-19 patients.

## 2. Methods

### 2.1. Study Design and Patient Cohort

The aim of this single-center observational study at the Klinikum Rechts der Isar of the Technical University of Munich was to investigate possible associations between alterations in intestinal and bronchial microbial composition and the occurrence of COVID-19-associated pulmonary aspergillosis (CAPA) in critically ill patients.

To this end, we examined a total of 42 patients treated in our ICU for COVID-19-induced respiratory or multi-organ failure between April and July 2020. SARS-CoV-2 infection was confirmed by quantitative reverse transcription PCR (TaqMan™-PCR performed on Roche cobas^®^ 6800, Basel, Switzerland), performed on nasopharyngeal swabs or tracheal fluid. During their ICU stay, patients were prospectively screened every three days for the development of CAPA using respiratory specimens. In addition to galactomannan testing from serum and respiratory samples, routine microbiological tests were performed at the same frequency. All patients received a computed tomography (CT) scan of the chest before ICU admission. Sixteen patients (38%) were diagnosed with CAPA at the time of study enrollment, based on the ECMM/ISHAM consensus criteria [14]. Every COVID-19 patient fulfilling these criteria was discussed by experts in microbiology, infectious disease, and intensive care medicine to ascertain that the criteria for CAPA were met. The remaining 26 patients served as CAPA-negative COVID-19 controls. The severity of COVID-19 disease was defined using the WHO Ordinal Scale for Clinical Improvement in Hospitalized Patients with COVID-19 [15]. All patients were seriously ill with an average WHO grade of 6.8 (standard deviation: 1.33), and some patients died in the course of the disease (*n* = 15). Patients received both parenteral and enteral feeding, the latter via a gastric tube (artificial feeding). Antibiotic therapies were classified according to their effect on the gut microbiome as described previously [13]. We evaluated the application of antibiotic therapy up to two weeks before samples for microbial analysis were taken. Neither vancomycin nor linezolid were administered in this cohort. Lastly, we noted antimycotic therapies in CAPA patients.

### 2.2. Microbial Sampling and 16S rRNA Gene Sequencing

We collected a total of 14 tracheal and 40 stool samples for microbial profiling from the 42 COVID-19 patients treated in our ICU. Microbial samples had matching complete blood count (CBC) and serum samples available. For intubated patients, the stool was collected either by a Flexi-Seal^TM^ (Convatec, Munich, Germany) device (*n* = 13), a fecal collector (stool collection bag, *n* = 23), or regular means (*n* = 4). Tracheal secretions were obtained by endobronchial aspiration or bronchoalveolar lavage, respectively, via their respiratory tube.

Preparation and sequencing of microbial samples were carried out as described previously [13]. Samples were stored in a solution to stabilize DNA (MaGix PBI, Microbiomix GmbH, Regensburg, Germany). Sample preparation and paired-end sequencing was performed on an Illumina MiSeq, targeting the V3V4 region of the 16S rRNA gene (using primers 341F and 785R). Raw FASTQ files were processed using the NGSToolkit (https://github.com/TUM-Core-Facility-Microbiome/ngstoolkit, accessed on 24 January 2022) based on USEARCH to generate denoised zero-radiation operational-taxonomic units (zOTUs). We excluded zOTUs with a relative abundance < 0.1% and a prevalence < 5%. Assessment of alpha diversity and taxonomic binning were computed using the Rhea software pipeline [16].

### 2.3. Statistical Analysis

Statistical analysis of 16S rRNA profiles was conducted as described previously [13]. Briefly, read counts were normalized and differences in the relative abundance of taxa and/or zOTUs were determined by a Kruskal–Wallis Rank Sum test for multiple groups and Mann–Whitney U tests for pairwise comparisons, respectively. Associations between categorical variables were tested using Fisher’s Exact test. The similarity between samples was estimated based on pairwise generalized UniFrac distances. Confounders and possible effect modifiers of microbial ecosystems were determined via a permutational multivariate analysis of variance using the aforementioned distance matrix.

### 2.4. Ethical Approval

Patients signed the informed consent either before intubation or after extubation. For deceased patients, the ethics committee permits the use of anonymized data. The institutional review board for human studies approved the protocols, and written consent was obtained from the subjects or their surrogates if required by the institutional review board. The study was conducted in accordance with the declaration of Helsinki and approved by the ethics committee of the Technical University Hospital of Munich (221/20 S-SR).

## 3. Results

### 3.1. CAPA Occurs More Frequently among Elderly COVID-19 Patients and Indicates an Unfavorable Clinical Course

Among our cohort of 42 COVID-19 patients selected for microbial sampling, 16 (38%) received a diagnosis of CAPA (Figure 1A). A basic summary of clinical characteristics and patient-level information is provided in Appendix A. We first examined clinical covariates for their association with CAPA status and only found age to differ significantly between groups. Specifically, the median age among COVID-19 patients with CAPA was 76.5 years as opposed to 61.5 years in non-CAPA patients (Figure 1B, *p* = 0.008, Mann–Whitney U test). Further trends of a higher occurrence in CAPA patients could be appreciated for age-related pre-existing conditions such as coronary artery disease and atrial fibrillation (Appendix A). Consistent with previous reports [17], COVID-19 patients with pulmonary *Aspergillus* superinfection exhibited a worse disease course, prolonged mechanical ventilation (Appendix A), and higher fatality rates (Figure 1C). Accordingly, vasopressor therapy was needed significantly more often in CAPA patients (Figure 1D). Importantly, the application of dexamethasone was balanced between CAPA and non-CAPA cases (Appendix A). Taken together, we confirmed that CAPA is associated with a significantly increased COVID-19 disease severity, and we found older age to be the main risk factor for contracting CAPA in our cohort.

### 3.2. Consideration of Factors Influencing the Microbiome and Comparability between Study Groups

16S rRNA gene amplicon sequencing of 40 fecal samples available from the 42 COVID-19 patients treated in our ICU revealed diverse ecosystems dominated by the two major phyla *Firmicutes* and *Bacteroidota* (cumulative mean relative abundance, 85%). Importantly, we did not detect significant differences in gut microbial composition between CAPA and non-CAPA patients (Figure 2A and Appendix A). Consequently, we evaluated further samples and patient metadata more carefully in an unbiased manner using a multivariate analysis of inter-individual variabilities in microbiota structure (Appendix A). Here, we found ‘days since hospital admission’ as the most prominent factor (*p* = 0.001) (Figure 2B). Both species richness and the abundance of a total of 47 genera correlated significantly (false discovery rate < 0.1) with days passed since hospital admission (Appendix A). Most genera showed a negative relationship, including *Actinomyces*, *Ruminococcus,* and *Bifidobacterium*. On the other hand, *Enteroccocus* and *Staphylococcus* genera represented exceptions with increasing relative abundance during the hospital stay.

Further influential covariates included ‘total length of ICU stay’, dexamethasone and antibiotic treatment, and whether patients were considered to be late in their COVID-19 disease course, i.e., if they had already cleared the virus from their upper airway as verified by PCR testing (Figure 2B). In contrast to the above covariates, we did not find feeding, stool collection type, or age to exhibit a significant influence on the global microbiota structure in these 40 stool samples.

Patients with late COVID-19 were both hospitalized and treated in the ICU for a significantly longer time than their early disease counterparts (Appendix A). Furthermore, dexamethasone treatment was absent in late disease patients, and fecal management more often involved a Flexi-Seal^TM^ device (Appendix A). A lack of antibiotic therapy tended to occur more often in patients with early COVID-19 disease. Lastly, stool microbial alpha diversity was significantly lower in patients with late COVID-19 disease (Appendix A).

For further investigation into microbial changes accompanying CAPA, we accounted for the aforementioned factors by stratifying patients both by early (ED) versus late disease (LD) and non-CAPA versus CAPA status yielding four groups: ED non-CAPA (*n* = 15), ED CAPA (*n* = 10), LD non-CAPA (*n* = 9) and LD CAPA (*n* = 6). With this approach, no significant difference in hospitalization length (Figure 2C), antibiotic therapy, feeding, or stool collection method could be observed between the respective CAPA and non-CAPA groups (Figure 2D). Expectedly, antimycotic therapy was only applied in CAPA patients with voriconazole and liposomal amphotericin B, respectively, dominating in early and late CAPA cases, respectively.

We further examined microbial samples from tracheal secretions, which were available from nine non-CAPA and five CAPA patients. Permutation analysis of inter-individual variabilities in microbiota structure found no significant role for CAPA status and again identified ‘days since admission’ as the most influential covariate. The latter was highest in four late disease COVID-19 patients. However, the limited patient number in this subset prohibited further statistically sound comparisons between non-CAPA and CAPA patients.

In summary, stool and bronchial microbial communities showed increased damage with the length of hospital stay. Adjustment for SARS-CoV-2 disease course status yielded groups of CAPA and non-CAPA patients that were balanced concerning major influences on stool inter-individual variabilities.

### 3.3. Impaired Fecal Microbiota and Staphylococcal Outgrowth Occur in Late but Not Early CAPA

Next, we analyzed the association of CAPA and gut microbial signatures in more detail. In the ED group, no difference in alpha diversity could be demonstrated for CAPA or non-CAPA patients (*p* = 0.89, Mann–Whitney U test). On the other hand, LD CAPA patients tended to show a decrease in microbial richness compared to LD non-CAPA cases (*p* = 0.11, Figure 3A). Similarly, LD CAPA samples were significantly overrepresented (Odds ratio = 15.0, *p* = 0.01 Fisher’s Exact test) among the least diverse 20 percent of samples as determined by sample richness (Figure 3B). No linear correlation could be observed for changes in gut microbial composition between CAPA and non-CAPA patients in ED versus LD, respectively, (Figure 3C), suggesting that pulmonary aspergillosis associates with distinct metagenomic shifts in each condition. Interestingly, we observed a higher abundance of *Staphylococcus epidermidis* in the stool of LD CAPA patients compared with LD non-CAPA patients. (*p*: 0.008, Mann–Whitney U test, Figure 3D). Thus, we found both a trend towards a reduced gut microbial within-sample diversity and a specific expansion in *S. epidermidis* relative abundance separating CAPA from non-CAPA cases late in COVID-19 disease.

## 4. Discussion

The COVID-19 global pandemic has been and still is impacting many aspects of our lives. Although vaccines have reduced morbidity, COVID-19 remains a threatening and serious disease, especially in predisposed and immunocompromised patients. Viral infections of the lung are prone to fungal superinfection [18], likely facilitated by inflammatory damage to the lung epithelium and an impaired immune response associated with the disease [18,19]. Coinfection, on the other hand, leads to a significant worsening of respiratory function and an increase in mortality for patients with viral pneumonia [20]. This also applies to SARS-CoV-2 infection, in which coinfection with *Aspergillus fumigatus* can lead to COVID-19-associated pulmonary aspergillosis with poor outcomes and the need for intensive care [5,7]. Risk factors for CAPA include severe lung injury in the setting of COVID-19, older age, and preexisting pulmonary disease [21]. In addition, the use of broad-spectrum antibiotics has been associated with the development of invasive pulmonary aspergillosis (IPA) [22], and the use of azithromycin also appears to predispose to CAPA [23]. While systemic corticosteroids are known risk factors for IPA [24], data on their role in increasing the risk for CAPA remain controversial [7,21]

The gut microbiome has a critical influence on the maturation of intestinal lymphoid immune defenses [25] and, moreover, on the immune response in peripheral tissues via multiple interactions with host immune cells [9,26,27]. Furthermore, there is evidence for an interrelation between intestinal bacteria and mycobiota [28] with the latter relying on bacterial metabolites such as short-chain fatty acids [29]. Studies in mice have shown that intestinal microbiota influence the adaptive immune response to *Aspergillus fumigatus* in the lung via the regulation of CD4 cells [30]. As has been noted before, it is plausible to assume that disturbances in the natural microbial composition of the gut and lung, respectively, can contribute to invasive aspergillosis [31]. However, studies on the relationship between CAPA and the gut microbiome are lacking.

Here, we were able to show for the first time that CAPA patients exhibit impaired stool and bronchial microbial communities late in COVID-19 disease. Specifically, late CAPA patients are overrepresented among stool samples with the lowest alpha diversity, and their fecal samples exhibit a significant increase in the abundance of *S. epidermidis.*

*S. epidermidis* belongs to the group of coagulase-negative staphylococci and is a facultative pathogen, known to be a root cause of nosocomial catheter and bloodstream infections. The bacterium furthermore serves as a genetic reservoir for *S. aureus,* increasing the pathogenicity and antibiotic resistance of *S. aureus* via gene transfer [32]. While *S. epidermidis* occurs mainly on the skin of adults, some strains appear to be well adapted to the gut ecological niche [33]. While studies on the prevalence and function of *S. epidermidis* in the adult gut are scarce, it has been shown that strains of *S. epidermidis* from the gut or skin essentially do not differ in their ability to form biofilms or pathogenicity [33]. A recent study has identified the gut as a reservoir for and origin of bloodstream infection with *S. epidermidis*, thus challenging the informal dogma that this infection originates exclusively from the environment of the skin [34]. Importantly, colonization of the intestine with *S. epidermidis* in mice leads to serious damage to various extraintestinal organs, such as the kidneys and the liver [35].

We note several limitations of our study. First, the limited number of cases precludes more robust conclusions at this time, and further studies with higher numbers of cases are needed. Second, while we considered the effect of several known confounders on gut microbial health such as antibiotic therapy and feeding type, the design of our study did not allow us to control for antimycotic therapy. As can be expected, only CAPA patients received either voriconazole or liposomal amphotericin B with the former being applied more often in early disease CAPA and the latter dominating in late disease CAPA patients. Thus, we cannot rule out conclusively that the differences in *S. epidermidis* abundance and alpha diversity we found in late-disease CAPA patients are direct or indirect effects of liposomal amphotericin B treatment, although we consider this to be unlikely. Reports on their influence on intestinal microbiome composition are scarce and lacking for antimycotic agents and amphotericin B, respectively. However, a recent study in mice noted no difference in fecal microbial species richness in animals treated with fluconazole as compared to controls [36]. Lastly, we would like to note that, at this time, we would not withhold therapy with liposomal amphotericin B in a patient with CAPA for the sake of preventing *S. epidermidis* outgrowth in the gut. Third, all samples were collected during the early phase of the pandemic when the SARS-CoV-2 wild-type variant dominated. Therefore, our results cannot be extrapolated to critically ill patients suffering from infections with more recent coronavirus variants without further study.

While the immunological consequences of the suggested decrease in stool microbial alpha diversity and outgrowth of coagulase-negative staphylococci remain uncertain, these findings certainly offer a therapeutic window via the application of probiotics, adaption of stool management or choice of antimycotic agent. These options and the evaluation of microbial ecosystems in patients should be continued in studies using larger case numbers.

## Figures and Tables

**Figure 1 jof-08-01265-f001:**
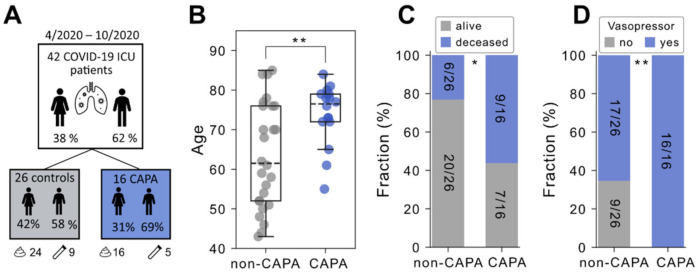
(**A**) Overview of the study design. Stool and tracheal secretions are indicated. (**B**) Distribution of age according to COVID-19-associated pulmonary aspergillosis (CAPA). The *p*-value is derived from a two-tailed Mann–Whitney U test. (**C**) Fraction of deceased patients according to CAPA status. (**D**) Fraction of patients requiring vasopressor therapy according to CAPA status. In boxplots, the box ranges from Q1 (the first quartile) to Q3 (the third quartile) of the distribution, and the range represents the IQR (interquartile range). The median is indicated by a dashed line across the box. The “whiskers” on box plots extend from Q1 and Q3 to 1.5 times the IQR. Unless otherwise specified, *p*-values are derived from a two-tailed Fisher’s Exact test. ** *p* ≤ 0.01; * *p* ≤ 0.05; ns, not significant.

**Figure 2 jof-08-01265-f002:**
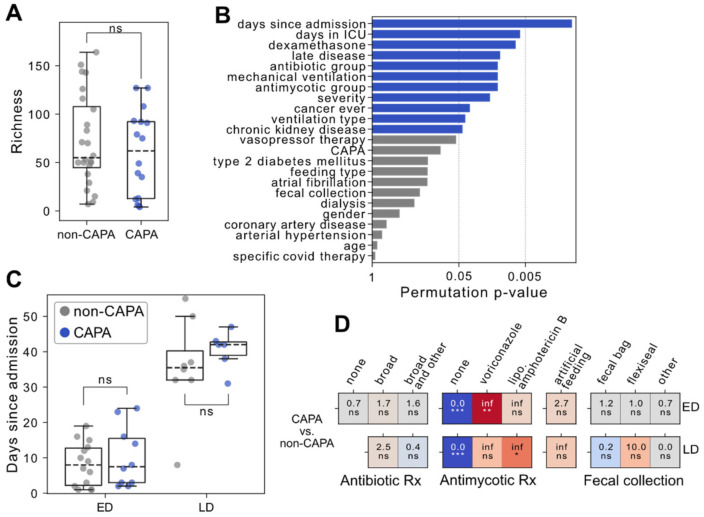
(**A**) Richness (=number of detected zOTU) on *y*-axis as a measure of sample diversity across stool samples from COVID-19 with and without CAPA. (**B**) *p*-values derived from a permutational multivariate analysis of variances (*x*-axis) for the indicated clinical covariates and the stool microbial ecosystem diversity as represented by generalized UniFrac distances. Blue bars mark significant covariates. (**C**) Days since admission to the hospital (*y*-axis) across patients with early vs. late COVID-19 disease and with or without *Aspergillus* superinfection (**D**) Association of antibiotic and antimycotic treatments, feeding type and stool collection method (*x*-axis) and early and late COVID-19 disease with *Aspergillus* infection (*y*-axis). For each combination and cell, the Odds ratio (OR, upper number) and a *p*-value code (from a two-sided Fisher’s exact test) are depicted. Cells are colored by *p*-value and the sign of log(OR). In boxplots, the box ranges from Q1 (the first quartile) to Q3 (the third quartile) of the distribution, and the range represents the IQR (interquartile range). The median is indicated by a dashed line across the box. The “whiskers” on box plots extend from Q1 and Q3 to 1.5 times the IQR. Unless otherwise specified, *p*-values are derived from two-tailed Mann–Whitney U tests. *** *p* ≤ 0.001; ** *p* ≤ 0.01; * *p* ≤ 0.05; ns, not significant.

**Figure 3 jof-08-01265-f003:**
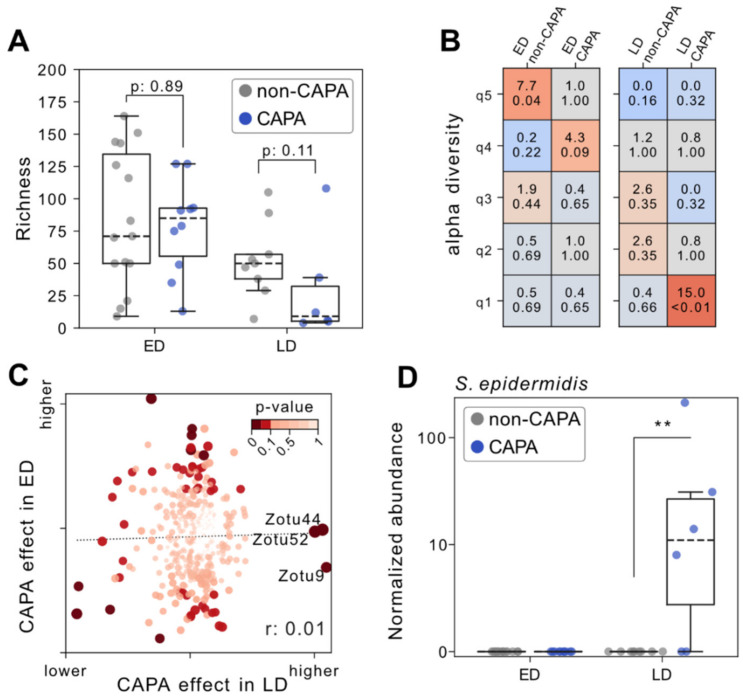
(**A**) Richness (=number of detected zOTU) on *y*-axis across patients grouped according to COVID-19 disease stage (ED, early disease vs. LD, late disease) and *Aspergillus* superinfection. (**B**) Association of microbial diversity quintile (q1 = lowest to q5 = highest) (*y*-axis) and patient groups according to early (ED) vs. late (LD) COVID-19 infection and presence of CAPA. For each combination and cell, the Odds ratio (OR, upper number) and *p*-value (from a two-sided Fisher’s exact test) are depicted. Cells are colored by *p*-value and sign of log(OR). (**C**) Scatter plot illustrating the effect of *Aspergillus* superinfection on microbial signatures (=difference between U2 and U1 from a Mann–Whitney U test) for stool specimens from late disease (*x*-axis) and early disease (*y*-axis) patients, respectively. Dots (=individual zOTUs) are colored by the minimum *p*-value from both comparisons as calculated from a two-sided Mann–Whitney U test. (**D**) Median normalized abundance on the *y*-axis from zOTUs belonging to S. epidermidis across patients grouped according to COVID-19 disease stage and *Aspergillus* superinfection. In boxplots, the box ranges from Q1 (the first quartile) to Q3 (the third quartile) of the distribution and the range represents the IQR (interquartile range). The median is indicated by a dashed line across the box. The “whiskers” on box plots extend from Q1 and Q3 to 1.5 times the IQR. Unless otherwise specified, *p*-values are derived from two-tailed Mann–Whitney U tests. ** *p* ≤ 0.01; ns, not significant.

## Data Availability

De-identified clinical annotations for each of the 42 patients are provided as Appendix A. FASTQ files of the 16S rRNA gene sequencing are available under SRA accession number PRJNA853783. Both data types will be made available without access restrictions upon publication.

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
