# Peer review of "Gut Microbial Disruption in Critically Ill Patients with COVID-19-Associated Pulmonary Aspergillosis"

_jof, 2022, doi:10.3390/jof8121265_

Round 1

Reviewer 1 Report

This elegant study shows that late in COVID-19 disease, CAPA patients can exhibit a trend towards reduced gut microbial diversity. Whether these alterations in the gut microbiome have influenced the development of severe CAPA or not cannot be confirmed based on the not-so-high number of patients enrolled. Nevertheless, this study demonstrates the complexity of COVID-19 pathophysiology and the need for further investigation.

Author Response

Point 1: English language and style are fine/minor spell check required

Response 1: Thank you very much for the time and thought you invested in reviewing our manuscript. We have edited the manuscript for readability as per your suggestion. 

Reviewer 2 Report

The authors attempt to correlate gut microbiome changes with the presence/absence of pulmonary Aspergillosis in COVID-19 patients. While the majority of the results presented are sound, one major confounder remains unaddressed. Were the patients on antifungal therapy? The use of other medications were accounted for - steroids and antibiotics in particular. Antibiotics are expected to influence microbial community structure. What is ignored throughout the manuscript is whether or not patients with the fungal infection were on antifungals and if so, how might the presence of that drug influenced the gut microbiome? It is completely possible that the changes reported are a result of other medications and not the presence of Aspergillus in the lung.  

Author Response

Point 1:  The authors attempt to correlate gut microbiome changes with the presence/absence of pulmonary Aspergillosis in COVID-19 patients. While the majority of the results presented are sound, one major confounder remains unaddressed. Were the patients on antifungal therapy? The use of other medications were accounted for - steroids and antibiotics in particular. Antibiotics are expected to influence microbial community structure. What is ignored throughout the manuscript is whether or not patients with the fungal infection were on antifungals and if so, how might the presence of that drug influenced the gut microbiome? It is completely possible that the changes reported are a result of other medications and not the presence of Aspergillus in the lung.  

Response 1: Thank you very much for your review of our manuscript and for raising the important issue of antifungal therapy in our cohort. 

As per your suggestion we have added information on antifungal therapy in our patient cohort. Out of 16 CAPA patients, 13 were in fact treated with an antimycotic agent. 8 patients were treated with voriconazole and 5 patients were treated with liposomal amphotericin B. 

We have added this information on a per-patient level to Supplementary Table 2. Furthermore, we have included antifungal therapy into our confounder analyses and have updated Figures 2B and 2D as well as Supplementary Tables 1 and 3 accordingly.

As can be expected, antifungal therapy is confounded with our main variable of interest, CAPA. We have stated this clearly in the updated Results section.  Furthermore, we find a preponderance of voriconazole therapy in early disease CAPA patients while late disease CAPA patients were treated more often with liposomal amphotericin B. This raises the possibility that our main results of a suggested decrease in alpha diversity and outgrowth of S. epidermidis in fecal samples of late disease CAPA patients could in part be due to the use of liposomal amphotericin B. 

We mention this specifically as a limitation of our study in the revised Discussion section. The literature on effects of antifungal medications, and particularly liposomal amphotericin B, on gut microbiota is scarce, to say the least. Studies investigating the effects other agents such as fluconazole did not find results similar to ours, that is decreased species richness and S. epidermidis outgrowth. 

Importantly, we have added a clinical opinion that we would not withhold antifungal therapy with liposomal amphotericin B in patients suffering from CAPA, be it in early or late disease for the sake of preventing S. epidermidis outgrowth or general gut microbial protection. 

Furthermore, we have edited the manuscript for readability.

We hope that our revised manuscript meets the requirements you specified in your review. Thank you again for the time and thought you put into reviewing our manuscript. 

Reviewer 3 Report

Very good paper. Congratulations.

Author Response

Thank you for the time and thought you put into reviewing our manuscript.